# The Effect of Hypertension on the Recovery of Renal Dysfunction following Reversal of Unilateral Ureteral Obstruction in the Rat

**DOI:** 10.3390/ijms24087365

**Published:** 2023-04-17

**Authors:** Fayez T. Hammad, Loay Lubbad, Suhail Al-Salam, Javed Yasin, Mohamed Fizur Nagoor Meeran, Shreesh Ojha, Waheed F. Hammad

**Affiliations:** 1Department of Surgery, College of Medicine & Health Sciences, United Arab Emirates University, Al Ain 15551, United Arab Emirates; 2Department of Pathology, College of Medicine & Health Sciences, United Arab Emirates University, Al Ain 15551, United Arab Emirates; 3Department of Internal Medicine, College of Medicine & Health Sciences, United Arab Emirates University, Al Ain 15551, United Arab Emirates; 4Department of Pharmacology and Therapeutics, College of Medicine & Health Sciences, United Arab Emirates University, Al Ain 15551, United Arab Emirates; 5School of Medicine, University of Jordan, Amman 11942, Jordan

**Keywords:** hypertension, renal functions, reversible unilateral ureteral obstruction

## Abstract

Both ureteral obstruction (UO) and hypertension are common conditions that affect kidney functions. Hypertension and chronic kidney disease are closely associated with an overlapping and intermingled cause-and-effect relationship. The effect of hypertension on the renal dysfunction following reversible UO has not been studied previously. To study this effect, spontaneously hypertensive (G-HT, *n* = 10) and normotensive Wistar (G-NT, *n* = 10) rats underwent 48-h reversible left unilateral UO (UUO), and the effect of UUO was studied 96 h following UUO reversal. The glomerular filtration rate, renal blood flow, and renal tubular functions such as the fractional excretion of sodium in the post-obstructed left kidney (POK) in both groups were significantly altered compared with the non-obstructed right kidney (NOK). However, the alterations in the G-HT were significantly more exaggerated when compared with the G-NT. Similar findings were observed with the histological features, gene expression of kidney injury markers, pro-inflammatory, pro-fibrotic and pro-apoptotic cytokines, and pro-collagen, as well as tissue levels of apoptotic markers. We conclude that hypertension has significantly exaggerated the alterations in renal functions and other parameters of renal injury associated with UUO.

## 1. Introduction

Ureteral obstruction is a common clinical problem that is caused by several conditions, such as urinary lithiasis. Following ureteral obstruction, several alterations occur in the renal functional parameters such as renal blood flow (RBF), glomerular filtration rate (GFR), and tubular functions [1,2,3,4]. Animal studies have also shown that following reversal of relatively short periods of ureteral obstruction, the main renal functional parameters, such as the RBF, GFR, urinary volume, and total and fractional excretion of sodium (FE_Na_), recover within a relatively short period of time [1].

Hypertension is a common disease worldwide. Hypertension and chronic kidney disease are closely associated with an overlapping and intermingled cause-and-effect relationship. Impairment in kidney functions is typically associated with an increase in blood pressure. On the other hand, long-term hypertension accelerates the deterioration of kidney functions [5,6,7,8,9]. Moreover, there is a direct relationship between the relative risk of developing end-stage kidney disease and hypertension severity [7,9]. In a large health screening registry, individuals with a baseline blood pressure of 180/100 mmHg were approximately 15 times more likely to develop end-stage kidney disease compared to individuals with a baseline blood pressure of 110/70 mmHg [9].

The relationship between ureteral obstruction and hypertension has been addressed by several studies [10,11,12,13,14]. However, the vast majority of these reports studied the effect of ureteral obstruction on the development or acceleration of hypertension. The impact of short periods of unilateral ureteral obstruction (UUO), the functions of the hypertensive kidney has not been investigated previously. Thus, the aim of this study was to examine this effect and compare it with that of the normal kidney in a UUO model in spontaneously hypertensive rats.

## 2. Results

The mean blood pressure before making UUO in G-NT and G-HT was 102 ± 9 and 176 ± 6, respectively (*p* = 0.0001). Just before the terminal experiment (after UUO), the blood pressure was 100 ± 8 and 170 ± 7, respectively (*p* = 0.001). However, during the terminal experiment under anesthesia, the mean arterial blood pressure and heart rate in the two groups were similar (96 ± 5 vs. 90 ± 5) and (312 ± 9 vs. 329 ± 13), respectively (*p* > 0.05 for both).

### 2.1. Glomerular and Tubular Functions

As shown in Table 1, the RBF and GFR in the POK in both G-NT and G-HT were significantly lower than the corresponding functions in the NOK. However, the effect of UUO on both parameters was more severe in G-HT, as demonstrated by comparing the percentage difference between the POK and NOK in each parameter in both groups (−107 ± 18 vs. −40 ± 11, *p* = 0.004 and −132 ± 13 vs. −55 ± 8, *p* = 0.001, respectively).

The FE_Na_ in the POK was significantly higher than the NOK in both groups (Table 1). Similar to the RBF and GFR, the percentage difference between the POK and NOK in the G-HT was higher than that in the G-NT, but this did not reach statistical significance (54 ± 16 vs. 21 ± 10, *p* = 0.08).

### 2.2. Urinary Albumin Creatinine Ratio

As shown in Figure 1, in the G-NT, the albumin creatinine ratio (ACR), post UUO reversal, was 23.8 ± 2.7 vs. 21.1 ± 1.7 in the pre-UUO value (*p* = 0.20). In the G-HT, however, the ACR following UUO-reversal was significantly higher than the pre-UUO ACR (67.7 ± 7.4 vs. 56.6 ± 8.7, *p* = 0.04).

### 2.3. Gene Expression Analysis Results

As demonstrated in Figure 2, in the G-HT, there was a 109.9 ± 31.0-fold increase in the expression of KIM-1 in the POK, whereas there was only an 18.3 ± 11.2-fold increase in the G-NT (*p* = 0.03). A similar trend was observed in the other marker of kidney injury, NGAL (5.4 ± 0.4 vs. 1.2 ± 0.2, *p* = 0.0001).

The changes in pro-inflammatory and pro-fibrotic cytokines and the pro-apoptotic p53 gene were similar to renal injury markers (Figure 3). The fold increase in the expression ratio of TNF-α, TGF-β1 and p53 in the POK compared with NOK in the G-HT and G-NT was (1.49 ± 0.23 vs. 0.91 ± 0.04 (*p* = 0.05), 1.83 ± 0.25 vs. 1.15 ± 0.08 (*p* = 0.045), and 1.35 ± 0.08 vs. 1.11 ± 0.05 (*p* = 0.03), respectively).

Similarly, there was a 4.57 ± 0.55-fold increase in the expression ratio of procollagen type-1 in the POK vs. NOK in the G-HT compared with only 2.67 ± 0.08 in the G-NT (*p* = 0.02) (Figure 4).

### 2.4. Western Blot Analysis

As shown in Figure 5, in G-NT, UUO caused a significant decrease in the expression of Bcl-2 and a significant increase in the BAX and BAX/Bcl-2 ratio. Similar trends were observed in G-HT; however, the changes in all these parameters were significantly more exaggerated in G-HT.

### 2.5. Histological Studies

In both groups (G-NT and G-HT), the right NOK had normal architecture and histology (no dilated tubules, tubular atrophy, or interstitial mononuclear cellular infiltrate; score 0 for all) (Figure 6 and Figure 7). The left POK in both groups showed significant histological abnormalities. In the G-NT, the POK showed foci of tubular dilatation (14.38 ± 2.51; score = 1), tubular atrophy (5.94 ± 0.64; score = 1) and interstitial mononuclear infiltrate (2.02 ± 0.25; score= 1) (Figure 6 and Figure 7). Similarly, the POK in the G-HT showed tubular dilatation (27.72 ± 4.21; score = 2), tubular atrophy (11.62± 2.30; score = 1) and an interstitial mononuclear infiltrate (2.08 ± 0.28; score = 1) (Figure 6 and Figure 7). The degree of both tubular dilatation and tubular atrophy in the POK in the G-HT was significantly greater than that observed in the POK in the G-NT (*p* = 0.01 and *p* = 0.02, respectively, Figure 7).

### 2.6. Gene Expression Analysis Results

As demonstrated in Figure 2, in G-HT, there was a 109.9 ± 31.0-fold increase in the expression of KIM-1 in the POK compared with NOK, whereas there was only an 18.3 ± 11.2-fold increase in the G-NT (*p* = 0.03). A similar trend was observed in the other marker of kidney injury, NGAL (5.4 ± 0.4 vs. 1.2 ± 0.2, *p* = 0.0001).

## 3. Discussion

In the current study, we have shown that hypertension has led to an exaggeration of the UUO-associated alterations in the renal hemodynamic and tubular functions and histological changes. It has also caused more pronounced alterations in the acute kidney injury markers and in pro-inflammatory, pro-fibrotic and pro-apoptotic cytokines, as well as the tissue level of the proapoptotic and antiapoptotic proteins.

Hypertension is a very common condition worldwide. It leads to alterations in the tone and reactivity of blood vessels, including those of the kidney. In an animal model, SHR were found to have higher renovascular resistance compared with normotensive Wistar rats [15] and this was found to be due to primary structural and functional abnormalities of the renal vessels [15]. The chronically raised blood pressure in the SHR has been shown to be accompanied by a lower GFR and RBF, reflecting the raised renal resistance [16]. The lower GFR and RBF in the G-HT compared with the G-NT in the current study are consistent with these findings.

The increased vascular resistance in hypertensive rats has been suggested to be due to either an increased production of or an increased sensitivity to vasoconstrictor substances such as vasoconstrictor prostaglandins, including thromboxane A2 [16,17,18]. In this regard, it has been shown that the renal production of prostaglandins is increased in SHR. [19]. This increased production and sensitivity to vasoconstrictor prostaglandins might explain the exaggerated deterioration in the GFR and PBF in the hypertensive rats compared with the normal rats in the current study because the production of vasoconstrictor prostaglandins such as thromboxane A2 has been shown to play an important role in the pathological alteration following ureteral obstruction. For instance, the production of thromboxane A2 has been shown to increase following ureteral obstruction, and pre-treatment with thromboxane synthetase inhibitors had a protective effect on the renal alterations associated with ureteral obstruction in rats [20,21].

In addition to the possible exaggerated response to the prostaglandins by the hypertensive animals, enhanced response to other vasoactive mediators might have also contributed to the exaggerated alterations observed in the response to the UUO in the hypertensive animals in this study. In hypertensive animals, the response of the renal vascular bed to norepinephrine, vasopressin, serotonin, and angiotensin II was shown to be enhanced and the degree of enhancement had increased with the duration of hypertension [15]. Certainly, many of these mediators especially the renin angiotensin system has been shown to play a major role in the renal alterations caused by UUO [22,23,24,25,26]. Whether the assumed exaggerated response of the renin angiotensin system occurred through stimulation of complement 3 [14] or other factors is difficult to ascertain from this study and further research is required to clarify this point.

This study has some other limitations. For instance, we did not measure the response of oxidative stress markers and other mediators such as dopamine and nitric oxide in both groups. These markers and mediators have been shown to be altered following UUO [27,28] and hence, it is possible that the exaggerated response of the SHR to the UUO could have been due to exaggerated alterations in these markers or mediators.

From the current data, it is difficult to ascertain if these exaggerated early alterations would be reflected on the renal functions in the long term, and longer-term studies are required to address this point. However, the increased gene expression of the pro-fibrotic cytokines and pro-collagen and of the BAX/Bcl 2 ratio observed in this study might indicate that even short periods of UUO (e.g., 48 h), which are usually followed by early complete recovery of the renal hemodynamic and tubular functions, might be associated with an exaggerated interstitial fibrosis in the long-term.

The current study addressed the changes in a model of hypertension that was not well controlled. Although it is likely that the response to UUO in a well-controlled model of hypertension would also be exaggerated, strictly speaking, the current study addressed hypertension, which is not well controlled. Further studies are required to investigate the UUO-associated renal alterations in a model of well-controlled hypertension.

The exaggerated response observed in hypertensive rats in response to UUO was similar to that observed in hypertensive animals with other acute renal injury conditions such as renal ischemia reperfusion injury [29]. Similarly, in a rat model, we have previously shown that diabetes mellitus led to a more pronounced response to short periods of UUO compared with normal animals [30]. Collectively, these data indicate that hypertension and other chronic conditions that affect the kidney lead to exaggerated alterations of the renal functions when the kidney is subjected to insults of various natures. However, these findings need to be validated in a human model and probably in a rat model using Wistar Kyoto (WKY) rats as controls.

In the current study, we have used Wistar rats rather than Wistar Kyoto (WKY) rats as controls for the SHR. Despite the fact that SHR and WKY rats are genetically related, WKY are a more heterogenous strain compared with SHR. In this regard, it appears that the distributed SHR have been fully inbred (i.e., after 20 generations of brother-sister mating), whereas the breeding stocks of WKY had been distributed before full inbreeding [31]. Accordingly, the biological variability of WKY may be greater than that of SHR. Kurtz and Morris demonstrated that WKY rats have significant differences in phenotypes (growth rate and blood pressure) depending on the commercial suppliers [32]. Furthermore, Rezende and colleagues studied the use of Wistar rats and Wistar WKY as controls for SHR by evaluating the blood pressure and cardiac structure and function [33]. They demonstrated that blood pressure values in WKY were intermediate between SHR and Wistar rats and close to the hypertension borderline. Wistar rats also showed pressure values that were more consistent with what was expected in normotensive rats. Moreover, WKY rats had earlier reductions in cardiac function when compared with Wistar rats. Due to these factors, some authors have actually used or suggested the use of Wistar rats as a control for SHR in several conditions [31,34,35,36].

In conclusion, hypertension led to an exaggeration of the renal functional alterations associated with UUO. It also caused more pronounced alterations in the histological features, acute kidney injury markers, and pro-inflammatory, pro-fibrotic and pro-apoptotic markers.

## 4. Materials and Methods

Studies were performed in male spontaneously hypertensive rats (SHR) and normotensive Wistar rats weighing 200–250 g at the time of ureteral occlusion. Rats were fasted for 12 h before the experimental procedures but had water ad libitum. The experimental protocol was approved by the local animal research ethics committee (ERA_2017_5691).

### 4.1. Experimental Groups

Animals were divided into two groups. G-NT (*n* = 10) were normotensive Wistar rats, whereas G-HT (*n* = 10) were spontaneously hypertensive rats. Both groups underwent left UUO for 48 h and terminal experiments for 96 h following the reversal of UUO.

### 4.2. Ureteral Occlusion and Reversal

Under aseptic conditions, the rats were anesthetized using intraperitoneal injections of ketamine hydrochloride (80 mg/kg, Pantex Holland B.V., Hapert, Holland) and xylazine hydrochloride (8 mg/kg, Troy Laboratory PTY Limited, NSW, Australia). As described previously [3,37,38], via a midline abdominal incision, the left ureter was exposed and obstructed by a 3–4 mm length of bisected PVC tubing (0.58 mm internal diameter), which was placed around the mid-ureter. The tube was then constricted with a 4-0 silk suture, and the wound was closed in layers.

Forty-eight hours later, the occlusion was reversed by removing the obstructing tube. Full release of the occlusion was confirmed by observing a free flow of urine across the occlusion site.

### 4.3. Surgical Procedure in the Terminal Experiment

Ninety-six hours following the release of UUO, the rats underwent terminal experiments to assess renal functions. Anesthesia was obtained using pentobarbital sodium (45 mg/kg, intraperitoneally; Sigma Life Science, St. Louis, MO, USA). As described previously [3], the trachea and the right femoral vein were cannulated, and the anesthesia was maintained by a continuous infusion of pentobarbital sodium (15 mg/kg/h). In addition, a sustaining infusion of 0.9% saline (50 µL/min) was started. The left femoral artery was cannulated with PE-50, and the tip of the cannula was positioned just below the level of the left renal artery for blood pressure measurement. Through a midline abdominal incision, both kidneys were exposed, and the upper ureters were cannulated with polyethylene tubing (PE-10) for urine collection. The sustaining infusion of 0.9% saline was then replaced by one composed of fluorescein isothiocyanate-inulin (FITC-inulin, Sigma-Aldrich, St. Louis, MO, USA) (2.5 mg/mL) and para-aminohippuric acid (PAH, Sigma-Aldrich, St. Louis, MO, USA) (0.4% *w*/*v*) in 0.9% saline. A priming dose of 2 mL of this solution was infused over 2 min. The rats were left for 75 min for equilibration before being subjected to the experimental protocol.

### 4.4. Experimental Protocol and Assays

The experimental protocol consists of two 20 min clearance periods. Arterial blood samples (0.4 mL) obtained at the beginning and end of each clearance period were immediately centrifuged, and the plasma samples (125 µL) were frozen. The plasma was then replaced by an equal volume of saline, and the erythrocytes were re-suspended by gentle vortexing and infused into the animal. The hematocrit was determined. Subsequently, the animals were euthanized, and the kidneys were removed, weighed, and prepared for gene expression analysis.

Flame photometry (Corning, Halstead, Essex, England) was used to determine the sodium level. The GFR and RBF were estimated from the clearances of inulin and PAH, respectively [3,37,38]. The values of GFR, RBF, and FE_Na_ were corrected for kidney weight.

### 4.5. Urine Collection and Measurement of Albumin Creatinine Ratio

Rats were placed individually in metabolic cages at various stages for urine collection and measurement of urine volume, urinary albumin, and creatinine level. Urine was collected for 24 h at the following points: one day before ureteral occlusion (baseline value) and on the 4th day following the reversal of the occlusion, i.e., one day prior to the terminal experiments.

### 4.6. Gene Expression Analysis

A wedge from the middle part of the kidney was excised, snap-frozen in liquid nitrogen, and stored at −80 °C for a later measurement of gene expression of the following:Markers of acute kidney injury: kidney injury molecule-1 (KIM-1) and neutrophil gelatinase-associated lipocalin (NGAL).Pro-inflammatory and pro-fibrotic cytokines: tumour necrosis factor-alpha (TNF-α) and transforming growth factor-β (TGF-β1).The pro-apoptotic gene p53.Procollagen type-1 (COL1A).

Total RNA was extracted from frozen samples as previously described [38]. The sequences of the primers and probes are shown in Table 2. The results were expressed as the mean fold change of gene expression in the POK compared with the NOK in each group.

### 4.7. Western Blot Analysis

Kidney tissues were homogenized in RIPA extraction buffer (Sigma Aldrich, St. Louis, MO, USA) containing a protease and phosphatase inhibitor cocktail (Thermo Fisher Scientific, Hanover Park, IL, USA) to get the total protein fraction. Kidney homogenates were centrifuged for 30 min at 18,000× *g* at 4 °C. Protein concentration in the kidney homogenates were estimated by the Pierce BCA Protein Assay (Thermo Fisher Scientific, Hanover Park, IL, USA). Equal quantities of the kidney protein samples were mixed with 4× Laemmli sample buffer (Bio Rad, Hercules, CA, USA) and 2-mercaptoethanol (Sigma Aldrich, St. Louis, MO, USA), then loaded and run on the SDS-PAGE. Separated protein samples were transferred onto PVDF membranes (Amersham Hybond P 0.45, PVDF, GE Healthcare Life Sciences, Solingen, Germany). Membrane blots were blocked with 5% skimmed milk in 1× TBST for one hour at room temperature. The blocked membranes were incubated overnight at 4 °C with the primary antibodies BAX (1:1000), Bcl-2 (1:2000), and GAPDH (1:5000). Blots were then incubated with the corresponding secondary antibodies for one hour at room temperature, and the protein bands were visualized with the SuperSignal West Pico PLUS Chemiluminescent Kit (Thermo Fisher Scientific, Hanover Park, IL, USA). The signal intensity (densitometry) of the developed bands was quantified using the ImageJ software (NIH, Bethesda, MD, USA).

Relative protein expression levels were calculated from the estimated densitometry values of two independent experiments after they were corrected with GAPDH expression to achieve equal protein loading.

### 4.8. Histological Studies

Kidneys were excised, washed with ice-cold saline, blotted with filter paper and weighed. The tissues from each kidney were cassetted and fixed directly in 10% neutral formalin for 24 h, which was subsequently dehydrated in increasing concentrations of ethanol, cleared with xylene, and embedded in paraffin. 3 μm sections were obtained from paraffin blocks and stained with hematoxylin and eosin. Evaluation of the stained sections was performed using light microscopy in a blind fashion.

Image J software (NIH, USA) was used to measure the frequency of observing the following histological abnormalities: tubular dilatation, tubular atrophy, interstitial fibrosis, and mononuclear cellular infiltrate. The microscopic scoring was performed by measuring the percentage of the areas showing morphologic abnormalities compared with the total surface area in each kidney sample. The following scoring system is used: (score 0: no abnormality), score 1: 1–25%, score 2: 26–50%, score 3: 51–75%, and score 4: 76–100%).

### 4.9. Statistical Analysis

Statistical analysis was performed using SPSS V16.0. Results were expressed as mean ± SEM. Because hypertension has already affected the NOK in G-HT, the percentage difference was used to study the effects of obstruction in the two groups. The percentage difference was calculated as the difference between the POK and NOK groups divided by the average of the two values multiplied by 100. A one-way factorial ANOVA was used for comparison of variables between the groups and between the POK and NOK within each group. The *p* value of less than 0.05 was considered statistically significant.

## Figures and Tables

**Figure 1 ijms-24-07365-f001:**
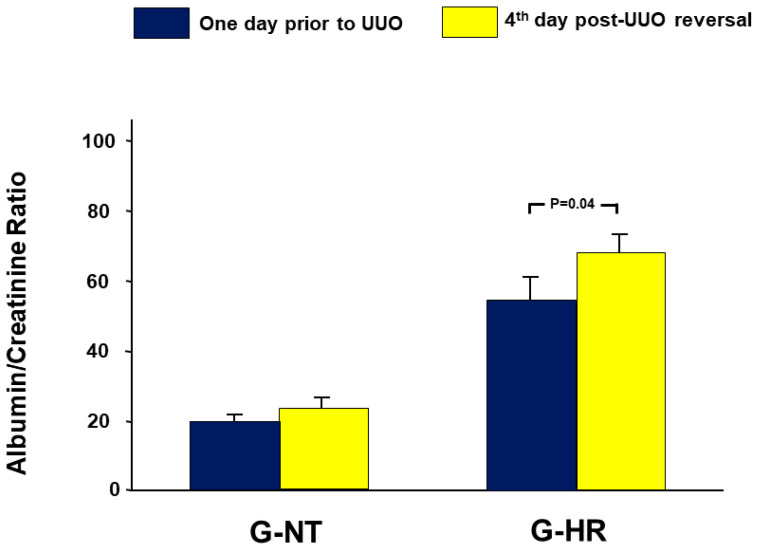
The albumin-creatinine ratio (ACR) one day prior to the creation of UUO (baseline value) and in the fourth day following the reversal of the UUO. Values represent the mean ± SEM.

**Figure 2 ijms-24-07365-f002:**
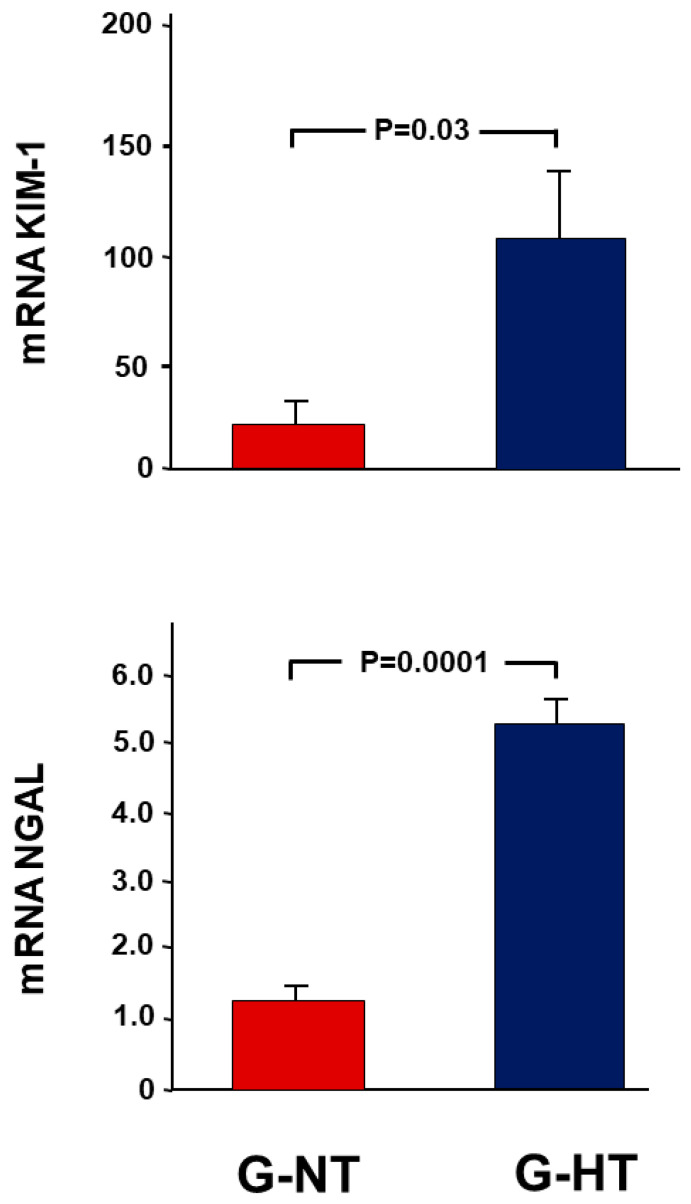
The fold-expression of the gene expression of two markers of acute renal injury (KIM-1 and NGAL) in the POK (post-obstructed kidney) compared with the NOK (non-obstructed control kidney) within each group. Values represent the mean ± SEM.

**Figure 3 ijms-24-07365-f003:**
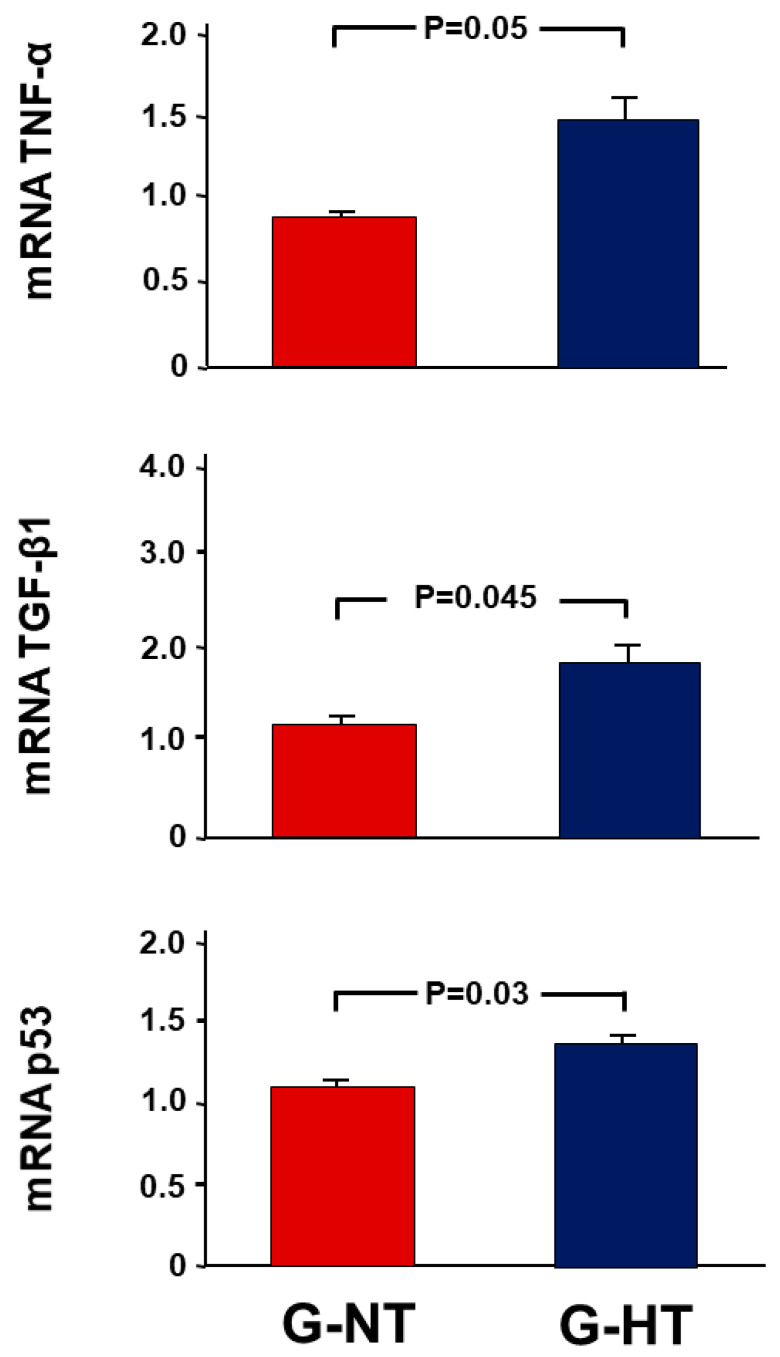
The fold-expression of the gene expression of pro-inflammatory (TNF-α), pro-fibrotic (TGF-β1) cytokines and pro-apoptotic gene p53 in the POK (post-obstructed kidney) compared with the NOK (non-obstructed control kidney) within each group. Values represent the mean ± SEM.

**Figure 4 ijms-24-07365-f004:**
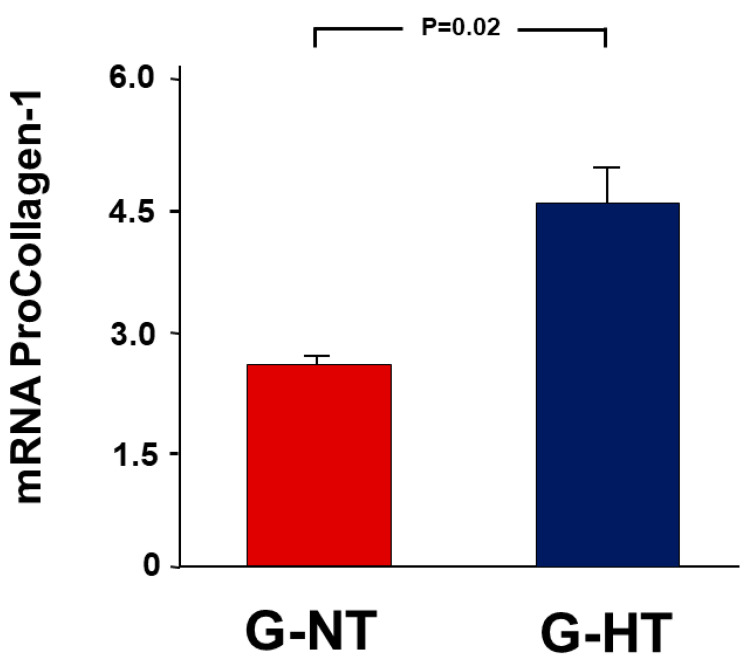
The fold-expression of the gene expression of the procollagen type-1 (COL1A) (G) in the POK (post-obstructed kidney) compared with the NOK (non-obstructed control kidney) within each group. Values represent the mean ± SEM.

**Figure 5 ijms-24-07365-f005:**
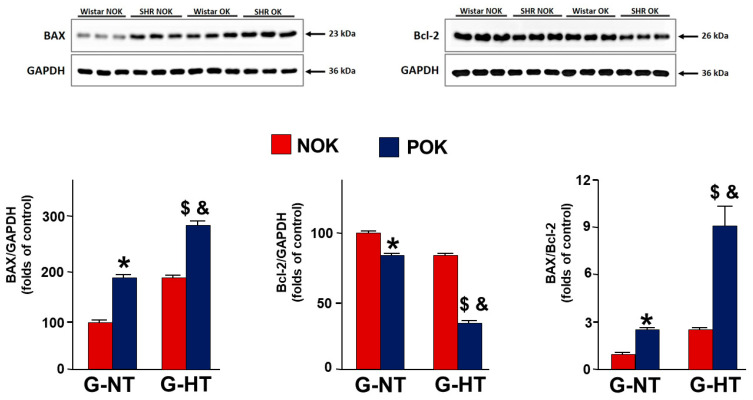
Western blot analysis showing the expression of BAX, Bcl-2, and the BAX/Bcl-2 ratio in the POK (post-obstructed kidney) and the NOK (non-obstructed control kidney) in the two groups. Values represent the mean ± SEM. * indicates statistical significance between the POK and NOK in G-NT; $ indicates statistical significance between the POK and NOK in G-HT; & indicates statistical significance in the % difference between the G-HT and G-NT rats.

**Figure 6 ijms-24-07365-f006:**
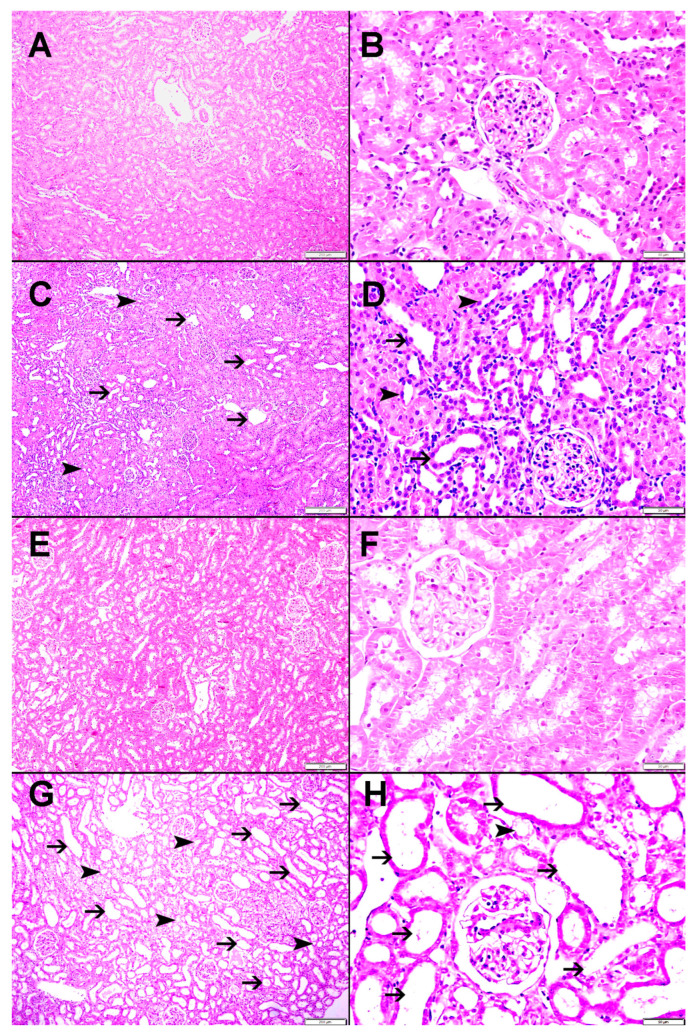
Histological features in the G-NT and G-HT groups. (**A**,**B**) the right NOK in the G-NT showing normal kidney architecture and histology with no dilated tubules, tubular atrophy, interstitial fibrosis, or interstitial mononuclear cellular infiltrate). (**C**,**D**) the left POK in the G-NT, which shows focal areas of tubular dilatation (thin arrow), and tubular atrophy (arrowhead). (**E**,**F**) the right NOK in the G-HT with normal kidney architecture and histology (no dilated tubules, tubular atrophy, interstitial fibrosis or interstitial mononuclear cellular infiltrate). (**G**,**H**) The left POK in the G-NT shows focal areas of tubular dilatation (thin arrow) and tubular atrophy (arrowhead). The white bar at the bottom right corner of each sub-figure represents a measurement scale. It measures 200 µm in (**A**,**C**,**E**,**G**) and 50 µm in (**B**,**D**,**F**,**H**).

**Figure 7 ijms-24-07365-f007:**
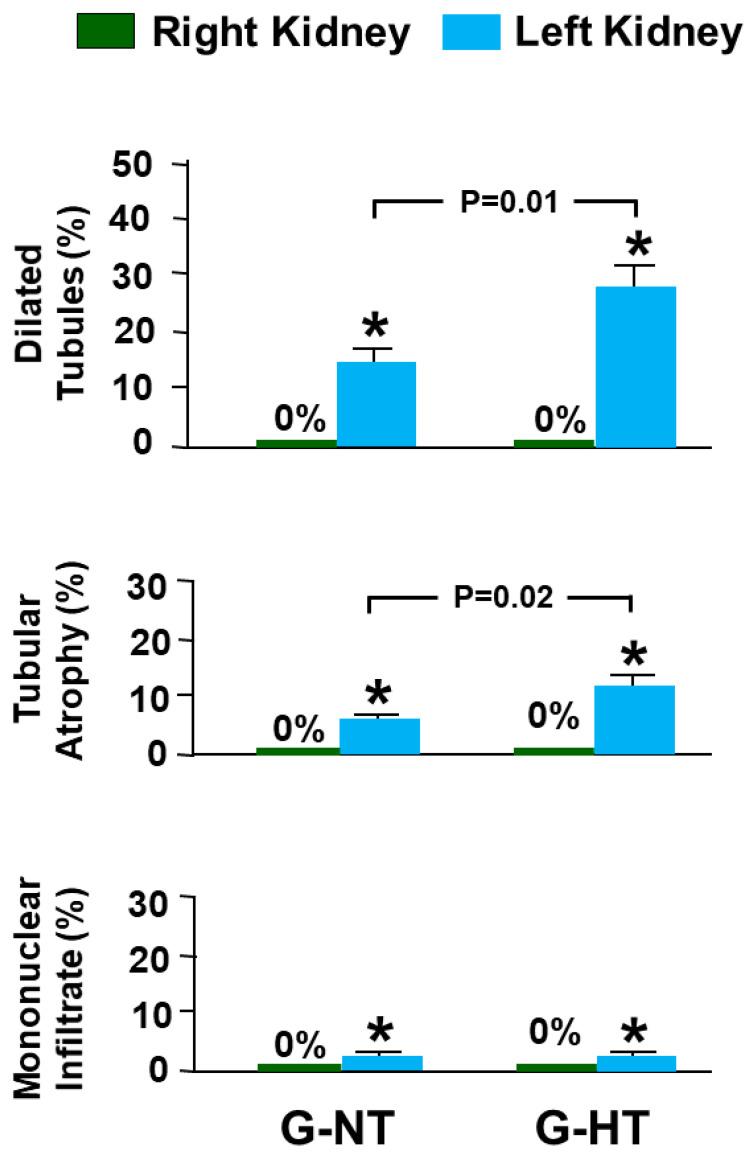
The score of various histological features in the left post-obstructed kidney (POK) and the right non-obstructed kidney in the G-NT and G-HT groups; s. Values represent mean ± SEM; * indicates statistical significance compared with right NOK in the same group.

**Table 1 ijms-24-07365-t001:** The glomerular filtration rate (GFR), renal blood flow (RBF), and the fractional excretion of sodium FENa in the left post-obstructed (POK) and right non-obstructed (NOK) kidneys in both the spontaneously hypertensive (G-HT) and the control normotensive (G-NT) rats.

	RBF			GFR			FE_Na_		
	NOK	POK	% Diff.	NOK	POK	% Diff.	NOK	POK	% Diff.
G-NT	5.98 ± 0.86	4.19 ± 0.69 *	−40 ± 11	0.95 ± 0.12	0.56 ± 0.09 *	−55 ± 8	0.031 ± 0.004	0.055 ± 0.010 *	21 ± 10
G-HT	3.88 ± 0.67 ^#^	1.08 ± 0.29 *	−107 ± 18 ^$^	0.48 ± 0.07 ^#^	0.11 ± 0.02 *	−132 ± 13 ^$^	0.027 ± 0.004	0.069 ± 0.013 *	54 ± 16

% Diff: the percentage difference between the POK and NOK in each group; * indicates statistical significance between the POK and NOK in each group; ^$^ indicates statistical significance in the % difference between the G-HT and G-NT rats; ^#^ indicates statistical significance between the NOK in the G-HT and G-NT rats.

**Table 2 ijms-24-07365-t002:** Forward and reverse primers as well as the fluorogenic probe sequences that were used in the real-time quantitative PCR analysis. KIM-1: kidney injury molecule-1; NGAL: neutrophil gelatinase-associated lipocalin, also called lipocalin 2 (Lcn2); TNF-α: tumour necrosis factor-alpha; TGF-β1: transforming growth factor-β; p53: the pro-apoptotic gene; and COL1A: procollagen type-1. PPIA: peptidylprolyl isomerase A (housekeeping gene).

Kim1 (NM_173149.2)	Forward	CTCACACTCAGATCATCTTCTC
	Reverse	CCGCTTGGTGGTTTGCTAC
	Probe	FAM-CTCGAGTGACAAGCCCGTAGCC-BHQ-1
NGAL (Lcn2) (NM_130741.1)	Forward	CTGTTCCCACCGACCAATGC
	Reverse	CCACTGCACATCCCAGTCA
	Probe	FAM-TGACAACTGAACAGACGGTGAGCG-BHQ-1
TNF-α (NM_012675.3)	Forward	CTCACACTCAGATCATCTTCTC
	Reverse	CCGCTTGGTGGTTTGCTAC
	Probe	FAM-CTCGAGTGACAAGCCCGTAGCC-BHQ-1
TGF-β1 (NM_021578.2)	Forward	GTGGCTGAACCAAGGAGACG
	Reverse	CGTGGAGTACATTATCTTTGCTGTC
	Probe	FAM-ACAGGGCTTTCGCTTCAGTGCTC-BHQ-1
p53 (NM_030989.3)	Forward	CGAGATGTTCCGAGAGCTGAATG
	Reverse	GTCTTCGGGTAGCTGGAGTG
	Probe	FAM-CCTTGGAATTAAAGGATGCCCGTGC-BHQ-1
COL1A (NM_053304.1)	Forward	CTGACTGGAAGAGCGGAGAGT
	Reverse	CCTGTCTCCATGTTGCAGTAGAC
	Probe	FAM-ACTGGATCGACCCTAACCAAGGC-BHQ-1
PPIA (NM_017101.1)	Forward	GCGTCTGCTTCGAGCTGT
	Reverse	CACCCTGGCACATGAATCC
	Probe	Quasar 670-TGCAGACAAAGTTCCAAAGACAGCA-BHQ-2

## Data Availability

The data presented in this study are available on request from the corresponding author.

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
