# Peer review of "The Effect of Hypertension on the Recovery of Renal Dysfunction following Reversal of Unilateral Ureteral Obstruction in the Rat"

_ijms, 2023, doi:10.3390/ijms24087365_

Round 1

Reviewer 1 Report

     The effect of hypertension on the recovery of kidney function following reversible ureteric obstruction is study in spontaneously hypertensive rats and control Wistar rats. The study is well performed and the results are clear; however, there are many studies on reversal ureteric obstruction in the setting of systemic illness.  The effects of diabetes, chronic renal disease, hypoxia or primary renal diseases increase the deleterious effects of ureteric obstruction than the observed in previously normal kidneys.

     In clinical medicine, physicians know that ureteral obstruction has deleterious effects on kidney function, and has to solved as soon as the diagnostic is performed. Then, the study is merely descriptive, evaluates some representative changes induced by tubular obstruction with the current available tools. 

1.-The proper control of spontaneously hypertensive rats (SHR) is Wistar Kyoto (WKY) rats; this strain should be used instead of Wistar rats, since WKY are genetically related to SHR.

2.- In order to evaluate the physiopathological mechanisms of hypertension on reversal ureteric obstruction, blood pressure should be controlled in SHR with drugs with low influence on kidney function and compared with normotensive controls (WKY) under reversible ureteric obstruction. 

Without these experiments results are not conclusive.

Author Response

Index of Changes- Manuscript ID: ijms-2206457

The effect of hypertension on the recovery of renal dysfunction
following reversal of unilateral ureteric obstruction in the rat

The authors would like to thank the Editor and the Reviewers for the positive critical comments which have strengthened the manuscript.

In this document, we have responded to the Reviewers’ comments. The changes refer to the highlighted revised version. In the highlighted version of the manuscript, the underlined text indicates that it has been added whereas strikethrough sign indicates deletion of the text. All modifications are indicated in red. A final neat version was also included.

In addition to addressing the Reviewer’s comments, we made further modifications in the text to render it easier to understand and follow by the reader (Page: 2, Line: 45-48), (Page: 3, Line: 61), (Page: 4, Line: 79), (Page: 5, Line: 93), (Page: 7, Line: 136 and 139-140), (Page: 8, Line: 163 and 174), (Page: 9, Line: 199), (Page: 10, Line: 200 and 202), (Page: 11, Line: 208, 210, 215-219, 224, 226 and 228), (Page: 12, Line: 238-239), (Page: 14, Line: 268, 272 and 278), (Page: 15, Line: 289-291), (Page: 16, Line: 310-311, 322-323, 325 and 328), (Page: 17, Line: 333), (Page: 18, Line: 355-356).

Reviewer#1:

The effect of hypertension on the recovery of kidney function following reversible ureteric obstruction is study in spontaneously hypertensive rats and control Wistar rats. The study is well performed and the results are clear; however, there are many studies on reversal ureteric obstruction in the setting of systemic illness.  The effects of diabetes, chronic renal disease, hypoxia or primary renal diseases increase the deleterious effects of ureteric obstruction than the observed in previously normal kidneys.

Comment #1: In clinical medicine, physicians know that ureteral obstruction has deleterious effects on kidney function, and has to solved as soon as the diagnostic is performed. Then, the study is merely descriptive, evaluates some representative changes induced by tubular obstruction with the current available tools.

Response: We would like to thank the Reviewer for the comment. Despite the fact that the results of this study make sense and probably anticipated, there was no previous data from the experimental animal models nor from human studies to support this assumption. In clinical practice, none of the urological guidelines including the guidelines of the major Urological Associations such as the American Urological Association and the European Association of Urology has documented these findings. For instance, if a patient presents with an acute ureteric obstruction due to ureteric stone, we tend to observe for up to 6 weeks before intervening surgically unless the patient had severe symptoms such as fever or severe uncontrollable pain regardless of the associated comorbidities such hypertension. Indeed, in patients with other comorbidities such as uncontrollable hypertension, we might sometimes delay intervention in a hope for spontaneous passage of the stone due to increased risk of anesthesia in such patients. Therefore, the current manuscript represents the first documented evidence of the effect of ureteric obstruction in a hypertensive model.

Comment #2: The proper control of spontaneously hypertensive rats (SHR) is Wistar Kyoto (WKY) rats; this strain should be used instead of Wistar rats, since WKY are genetically related to SHR.

Response: We would like to thank the Reviewer for this important point. In studies in which there is a difference between baseline values between the groups, it is better to use the percentage difference in the variables to compare the groups. In this study and due to the obvious differences in the baseline values between hypertensive and non-hypertensive groups, we have used the percentage difference in the variables rather than the absolute values to compare the two groups (Page: 9, Line: 198-199 and Page: 10, Line: 200 and 202). This renders it less important to use the Wistar rats or the Wistar Kyoto (WKY) rats as a control. Moreover, despite the fact that SHR and WKY are genetically related, WKY is a more heterogenous strain compared to SHR. In this regard, it appears that whereas the National Institutes of Health has distributed breeding stocks of SHR after they had been fully inbred (i.e., after 20 generations of brother-sister mating), the breeding stocks of WKY had been distributed before they were fully inbred. Accordingly, the biological variability of WKY may be greater than that of SHR [1]. Kurtz and Morris demonstrated that WKY rats have significant differences in phenotypes (growth rate and blood pressure) depending on the commercial suppliers [2]. Furthermore, Rezende and colleagues studied the use of Wistar rats and Wistar WKY as controls for SHR by assessing the long-term behavior of blood pressure and cardiac structure and function [3]. They demonstrated that blood pressure values in WKY were intermediate between SHR and Wistar rats and close to hypertension borderline. Wistar rats also showed pressure values that were more consistent with what was expected in normotensive rats. Moreover, WKY had earlier reductions in cardiac function when compared to Wistar rats. Due to these factors some authors have actually used or suggested the use of Wistar rats as control for SHR in several conditions [1, 4-6]. An extra text has been added to clarify this point (Page: 17, Line: 335-351).

Comment #3: In order to evaluate the physiopathological mechanisms of hypertension on reversal ureteric obstruction, blood pressure should be controlled in SHR with drugs with low influence on kidney function and compared with normotensive controls (WKY) under reversible ureteric obstruction.

Without these experiments results are not conclusive

Response: We agree with the Reviewer that in clinical practice, we have two types of hypertensive patients: those in which hypertension is well-controlled and those with uncontrolled disease. The current study addressed the changes in a model of hypertension which was not well-controlled. Although it is likely that the response of subjects with well-controlled hypertension to ureteric obstruction would also be exaggerated, strictly speaking, this study addressed hypertension which is not well-controlled. An extra-text has been added to the Discussion to address this important point (Page: 16, Line: 316-321).

References:

  1. Rocha, N.N., Are Wistar Rats the Most Suitable Normotensive Controls for Spontaneously Hypertensive Rats to Assess Blood Pressure and Cardiac Structure and Function? Int J Cardiovasc Sci, 2022. 35(2): p. 172-173.
  2. Kurtz, T.W. and R.C. Morris, Jr., Biological variability in Wistar-Kyoto rats. Implications for research with the spontaneously hypertensive rat. Hypertension, 1987. 10(1): p. 127-31.
  3. Rezende, L.M.T., et al., Is the Wistar Rat a more Suitable Normotensive Control for SHR to Test Blood Pressure and Cardiac Structure and Function? Int J Cardiovasc Sci, 2022. 35(2): p. 161-171.
  4. Moreira, N.J.D., et al., Acute renal denervation normalizes aortic function and decreases blood pressure in spontaneously hypertensive rats. Sci Rep, 2020. 10(1): p. 21826.
  5. Nakamura, A. and E.J. Johns, Renal nerves, renin, and angiotensinogen gene expression in spontaneously hypertensive rats. Hypertension, 1995. 25(4 Pt 1): p. 581-6.
  6. Shi, W., et al., Protective effect of calcitriol on podocytes in spontaneously hypertensive rat. J Chin Med Assoc, 2018. 81(8): p. 691-698.

Reviewer 2 Report

General comments:  This report deals with the effect of two days of unilateral ureteral obstruction (UUO) in normotensive and hypertensive rats.  This report is important because most studies related to UUO are performed after 7 days of UUO. There are also reports of studies after two days of UUO and three days of UUO. Two days of UUO increased ureteral inflammation (Eur Urol Focus. 2022 Oct 13;S2405-4569(22)00222-X. doi: 10.1016/j.euf.2022.09.014). The expression of miR-21, a microRNA associated with inflammation, was increased in the kidneys after 3 days of UUO (J Ethnopharmacol. 2022 Dec 10;304:115928. doi: 10.1016/j.jep.2022.115928). These studies, however, were performed in normotensive mice or rats; only a few studies on UUO have been performed in hypertensive rats (vide infra). The inclusion of studies following reversal of UUO adds strength to this report.

The authors discussed well the involvement of vasoconstrictors in UUO.  The authors may also want to include a reference on UUO and reactive oxygen (Front Pharmacol. 2022 Jan 7;12:798381. doi: 10.3389/fphar.2021.798381) and complement ( Am J Physiol Renal Physiol. 2013 Oct 1;305(7):F957-67. doi: 10.1152/ajprenal.00344.2013). A role of dopamine and nitric oxide in UUO has also been reported (Am J Physiol Renal Physiol. 2004 Mar;286(3):F509-15. doi: 10.1152/ajprenal.00253.2002).

 Specific concerns:

A.    Major:

1.     The mRNA but not protein expressions of KIM-1, NGAL, TNF-a, TGF-b1, p53, and ProCollagen-1 are shown in Figures 2-4.  

 B.    Minor:

1.     Page 2, lines 54 and 55. There are a few reports on the effect of “short periods” (depending on the definition of short-term) of UUO on the hypertensive kidney. “Unilateral ureteral obstruction (UUO) in 6-week old male spontaneously hypertensive rats (6-w-SHR) accelerated the elevation of blood pressure and developed stroke with high frequency from 3 weeks after operation, whereas UUO had no effect in either 20-week old SHR or 6-week old normotensive Wistar Kyoto rats” (Clin Exp Hypertens. 1980;2(1):139-52..doi: 10.3109/10641968009038557). Other reports related to UUO and hypertension include: (1) Am J Physiol. 1990;258(6 Pt 2):F1479-89. doi: 10.1152/ajprenal.1990.258.6.F1479; (2) Jpn Heart J. 1979;20(5):711. doi: 10.1536/ihj.20.711; (3) Acta Physiol (Oxf). 2007;189(1):67-75. doi: 10.1111/j.1748-1716.2006.01625; (4) J Urol. 2011;186(3):1142-9. doi: 10.1016/j.juro.2011.04.108; (5) J Urol. 2013 Mar;189(3):960-5doi: 10.1016/j.juro.2012.08.242; and Am J Physiol Renal Physiol. 2013;305(7):F957-67. doi: 10.1152/ajprenal.00344.2013, among others.

2.     Table 1. It seems that the % difference in FENa between G-NT and G-HT is significant.

3.     The manuscript needs to be edited for grammar and syntax. Some examples are listed.

a.     Page 1, line 16; page 2, line 37, etc. The word “ureteral” may be more appropriate than “ureteric” in the context of the statement.

b.     Page 2. lines 49-51. The format of end-stage kidney disease should be consistent.

c.      Page 2, line 60.  It may be better to change “creating” to “generating”. To create is to make something out of nothing. To generate is to do something and have an effect. (https://hinative.com/questions/2219.

d.     Page 2, line 67, page 3, line 104, page 4, line 111, page 5, lines 127 and 130, and page 13, line 335, among others. Change “to” to “with”. “To compare to is to point out or imply resemblances between objects regarded as essentially of a different order; to compare with is mainly to point out differences between objects regarded as essentially of the same order.” (https://www.dailywritingtips.com/compared-to-or-compared-with).

e.     Page 2, line 94 and title of the ordinate in Figure 1. Urinary albumin/creatinine is a ratio.  It may be better to write “Urinary albumin to creatine ratio” or “albumin creatinine ratio as in the legend of Figure 1 ([age 2, line 100).

f.       Page 4, line 110, page 5, line 126. There should be a hyphen between fold and expression as in fold-expression.

g. Page 6. The increased expression of BAX in G-NT and G-HT is not described in the text (page 6, lines 143-146).

Author Response

Index of Changes- Manuscript ID: ijms-2206457

The effect of hypertension on the recovery of renal dysfunction
following reversal of unilateral ureteric obstruction in the rat

The authors would like to thank the Editor and the Reviewers for the positive critical comments which have strengthened the manuscript.

In this document, we have responded to the Reviewers’ comments. The changes refer to the highlighted revised version. In the highlighted version of the manuscript, the underlined text indicates that it has been added whereas strikethrough sign indicates deletion of the text. All modifications are indicated in red. A final neat version was also included.

In addition to addressing the Reviewer’s comments, we made further modifications in the text to render it easier to understand and follow by the reader (Page: 2, Line: 45-48), (Page: 3, Line: 61), (Page: 4, Line: 79), (Page: 5, Line: 93), (Page: 7, Line: 136 and 139-140), (Page: 8, Line: 163 and 174), (Page: 9, Line: 199), (Page: 10, Line: 200 and 202), (Page: 11, Line: 208, 210, 215-219, 224, 226 and 228), (Page: 12, Line: 238-239), (Page: 14, Line: 268, 272 and 278), (Page: 15, Line: 289-291), (Page: 16, Line: 310-311, 322-323, 325 and 328), (Page: 17, Line: 333), (Page: 18, Line: 355-356).

Reviewer #2:

Comment #1: General comments: This report deals with the effect of two days of unilateral ureteral obstruction (UUO) in normotensive and hypertensive rats. This report is important because most studies related to UUO are performed after 7 days of UUO. There are also reports of studies after two days of UUO and three days of UUO. Two days of UUO increased ureteral inflammation (Eur Urol Focus. 2022 Oct 13;S2405-4569(22)00222-X. doi: 10.1016/j.euf.2022.09.014). The expression of miR-21, a microRNA associated with inflammation, was increased in the kidneys after 3 days of UUO (J Ethnopharmacol. 2022 Dec 10;304:115928. doi: 10.1016/j.jep.2022.115928). These studies, however, were performed in normotensive mice or rats; only a few studies on UUO have been performed in hypertensive rats (vide infra). The inclusion of studies following reversal of UUO adds strength to this report.

The authors discussed well the involvement of vasoconstrictors in UUO.  The authors may also want to include a reference on UUO and reactive oxygen (Front Pharmacol. 2022 Jan 7;12:798381. doi: 10.3389/fphar.2021.798381) and complement (Am J Physiol Renal Physiol. 2013 Oct 1;305(7): F957-67. doi: 10.1152/ajprenal.00344.2013). A role of dopamine and nitric oxide in UUO has also been reported (Am J Physiol Renal Physiol. 2004 Mar;286(3): F509-15. doi: 10.1152/ajprenal.00253.2002).

Response: We would like to thank the Reviewer for these important points. These have now been addressed and the text has been modified and an extra-text was added accordingly (Page: 15, Line: 269-305), (Page: 16, Line: 306-307).

Comment #2: The mRNA but not protein expressions of KIM-1, NGAL, TNF-a, TGF-b1, p53, and ProCollagen-1 are shown in Figures 2-4

Response: Indeed, and this what was mentioned in the Results section and in the Figures legends (Page: 12, Line: 229-240) and Figure 2-4.

Comment #3: Page 2, lines 54 and 55. There are a few reports on the effect of “short periods” (depending on the definition of short-term) of UUO on the hypertensive kidney. “Unilateral ureteral obstruction (UUO) in 6-week old male spontaneously hypertensive rats (6-w-SHR) accelerated the elevation of blood pressure and developed stroke with high frequency from 3 weeks after operation, whereas UUO had no effect in either 20-week old SHR or 6-week old normotensive Wistar Kyoto rats” (Clin Exp Hypertens. 1980;2(1):139-52..doi: 10.3109/10641968009038557). Other reports related to UUO and hypertension include: (1) Am J Physiol. 1990;258(6 Pt 2):F1479-89. doi: 10.1152/ajprenal.1990.258.6.F1479; (2) Jpn Heart J. 1979;20(5):711. doi: 10.1536/ihj.20.711; (3) Acta Physiol (Oxf). 2007;189(1):67-75. doi: 10.1111/j.1748-1716.2006.01625; (4) J Urol. 2011;186(3):1142-9. doi: 10.1016/j.juro.2011.04.108; (5) J Urol. 2013 Mar;189(3):960-5doi: 10.1016/j.juro.2012.08.242; and Am J Physiol Renal Physiol. 2013;305(7):F957-67. doi: 10.1152/ajprenal.00344.2013, among others

Response: The text has been modified and an extra text has been added to address this point (Page: 3, Line: 72-77) and (Page: 4, Line: 78-79).

Comment #4: Table 1. It seems that the % difference in FENa between G-NT and G-HT is significant

Response: We would like to thank the Reviewer for this very valid observation. We have re-checked the statistical analysis and it is still the same significance. This is due to the relatively large SEM compared to the mean value.

Comment #5: The manuscript needs to be edited for grammar and syntax. Some examples are listed.

  1. Page 1, line 16; page 2, line 37, etc. The word “ureteral” may be more appropriate than “ureteric” in the context of the statement

Response: The word ‘ureteric’ has been replaced with ‘ureteral, throughout the text (Page: 1, Line: 2), (Page: 2, Line: 31), (Page: 3, Line: 55, 56 and 59), (Page: 4, Line: 78-79), (Page: 14, Line: 283-285 and 287) and (Page: 17, Line: 330-331).

In addition, we made further modifications in the text to render it easier to understand and follow by the reader (Page: 2, Line: 45-48), (Page: 3, Line: 61), (Page: 4, Line: 79), (Page: 5, Line: 93), (Page: 7, Line: 136 and 139-140), (Page: 8, Line: 163 and 174), (Page: 9, Line: 199), (Page: 10, Line: 200 and 202), (Page: 11, Line: 208, 210, 215-219, 224, 226 and 228), (Page: 12, Line: 238-239), (Page: 14, Line: 268, 272 and 278), (Page: 15, Line: 289-291), (Page: 16, Line: 310-311, 322-323, 325 and 328), (Page: 17, Line: 333), (Page: 18, Line: 355-356).

Comment #6: Page 2. lines 49-51. The format of end-stage kidney disease should be consistent

Response: This has now been changed to be consistent (Page: 3, Line: 70).

Comment #7: Page 2, line 60.  It may be better to change “creating” to “generating”. To create is to make something out of nothing. To generate is to do something and have an effect. (https://hinative.com/questions/2219

Response: This has now been changed to avoid the word created (Page: 11, Line: 207).

Comment #8: Page 2, line 67, page 3, line 104, page 4, line 111, page 5, lines 127 and 130, and page 13, line 335, among others. Change “to” to “with”. “To compare to is to point out or imply resemblances between objects regarded as essentially of a different order; to compare with is mainly to point out differences between objects regarded as essentially of the same order.” (https://www.dailywritingtips.com/compared-to-or-compared-with).

Response: We would like to thank the Reviewer for this nice point. Changes were made as suggested (Page: 2, Line: 41 and 42), (Page: 4, Line: 81), (Page: 8, Line: 160), (Page: 9, Line: 191), (Page: 12, Line: 230, 235 and 239), (Page: 14, Line: 269, 273 and 281). (Page: 16, Line: 326), (Page: 17, Line: 332-333) and Figure-2, Figure-3, Figure-4 and Figure-7 legends.

Comment #9: Page 2, line 94 and title of the ordinate in Figure 1. Urinary albumin/creatinine is a ratio.  It may be better to write “Urinary albumin to creatine ratio” or “albumin creatinine ratio as in the legend of Figure 1 ([age 2, line 100).

Response: This has now been changed as per the suggestion (Page: 7, Line: 138) and (Page: 11, Line: 223-224).

Comment #10: Page 4, line 110, page 5, line 126. There should be a hyphen between fold and expression as in fold-expression

Response: This has been changed accordingly Figure-2, Figure-3 and Figure-4 legends.

Comment #11: Page 6. The increased expression of BAX in G-NT and G-HT is not described in the text (page 6, lines 143-146).

Response: The text was changed to address this point (Page: 12, Line: 243-246).

Round 2

Reviewer 1 Report

The authors did not answer my major concerns.

Author Response

Reviewer#1:

Comment #1: The authors did not answer my major concerns.

Response: We are not sure if the Reviewer had an access to our previous response which addressed all the Reviewer’s three points in detail and the necessary changes which were made accordingly. Our previous responses are shown below. In summary, in relation to the effect of ureteral obstruction on the kidney functions, although this fact is well-know, the effect of hypertension on the degree of this dysfunction was not known previously. Furthermore, in patients with uncontrolled hypertension, we sometimes tend to delay intervention due to fitness to anesthesia issues.

In relation to the use of Wistar rats as a control, some authors have recommended the use of Wistar instead of WKY rats as a control due to the reasons mentioned below. Furthermore, and due to different baseline values even with WKY rats, the comparison between groups should be performed using percentage differences and not absolute values; hence it is totally acceptable to use either WKY or Wistar rats as a control. However, we have added some further text (in addition to what was added previously) to address the potential benefit of using WKY in this model (Page: 14, Line: 398-400), (Page: 15, Line: 401-402).

Finally, with regard to the control of blood pressure, we indeed, controlled the blood pressure in the terminal experiment and during the measurement of kidney functions. During terminal experiment, the blood pressure in the two groups was similar. Prior to the terminal experiment, the hypertension was not controlled in the G-HT group and hence, this study addressed the dysfunction in the subgroup of subjects with uncontrolled hypertension.

The text modifications in relation to the above changes are shown in our previous responses (vide infra).

Previous Responses:

Comment #1: In clinical medicine, physicians know that ureteral obstruction has deleterious effects on kidney function, and has to solved as soon as the diagnostic is performed. Then, the study is merely descriptive, evaluates some representative changes induced by tubular obstruction with the current available tools.

Response: We would like to thank the Reviewer for the comment. Despite the fact that the results of this study make sense and probably anticipated, there was no previous data from the experimental animal models nor from human studies to support this assumption. In clinical practice, none of the urological guidelines including the guidelines of the major Urological Associations such as the American Urological Association and the European Association of Urology has documented these findings. For instance, if a patient presents with an acute ureteric obstruction due to ureteric stone, we tend to observe for up to 6 weeks before intervening surgically unless the patient had severe symptoms such as fever or severe uncontrollable pain regardless of the associated comorbidities such hypertension. Indeed, in patients with other comorbidities such as uncontrollable hypertension, we might sometimes delay intervention in a hope for spontaneous passage of the stone due to increased risk of anesthesia in such patients. Therefore, the current manuscript represents the first documented evidence of the effect of ureteric obstruction in a hypertensive model.

Comment #2: The proper control of spontaneously hypertensive rats (SHR) is Wistar Kyoto (WKY) rats; this strain should be used instead of Wistar rats, since WKY are genetically related to SHR.

Response: We would like to thank the Reviewer for this important point. In studies in which there is a difference between baseline values between the groups, it is better to use the percentage difference in the variables to compare the groups. In this study and due to the obvious differences in the baseline values between hypertensive and non-hypertensive groups, we have used the percentage difference in the variables rather than the absolute values to compare the two groups (Page: 13, Line: 325-329). This renders it less important to use the Wistar rats or the Wistar Kyoto (WKY) rats as a control. Moreover, despite the fact that SHR and WKY are genetically related, WKY is a more heterogenous strain compared to SHR. In this regard, it appears that whereas the National Institutes of Health has distributed breeding stocks of SHR after they had been fully inbred (i.e., after 20 generations of brother-sister mating), the breeding stocks of WKY had been distributed before they were fully inbred. Accordingly, the biological variability of WKY may be greater than that of SHR [1]. Kurtz and Morris demonstrated that WKY rats have significant differences in phenotypes (growth rate and blood pressure) depending on the commercial suppliers [2]. Furthermore, Rezende and colleagues studied the use of Wistar rats and Wistar WKY as controls for SHR by assessing the long-term behavior of blood pressure and cardiac structure and function [3]. They demonstrated that blood pressure values in WKY were intermediate between SHR and Wistar rats and close to hypertension borderline. Wistar rats also showed pressure values that were more consistent with what was expected in normotensive rats. Moreover, WKY had earlier reductions in cardiac function when compared to Wistar rats. Due to these factors some authors have actually used or suggested the use of Wistar rats as control for SHR in several conditions [1, 4-6]. An extra text has been added to clarify this point (Page: 14, Line: 398-400), (Page: 15, Line: 403-419).

Comment #3: In order to evaluate the physiopathological mechanisms of hypertension on reversal ureteric obstruction, blood pressure should be controlled in SHR with drugs with low influence on kidney function and compared with normotensive controls (WKY) under reversible ureteric obstruction.

Without these experiments results are not conclusive

Response: We agree with the Reviewer that in clinical practice, we have two types of hypertensive patients: those in which hypertension is well-controlled and those with uncontrolled disease. The current study addressed the changes in a model of hypertension which was not well-controlled. Although it is likely that the response of subjects with well-controlled hypertension to ureteric obstruction would also be exaggerated, strictly speaking, this study addressed hypertension which is not well-controlled. An extra-text has been added to the Discussion to address this important point (Page: 15, Line: 384-388).

References:

  1. Rocha, N.N., Are Wistar Rats the Most Suitable Normotensive Controls for Spontaneously Hypertensive Rats to Assess Blood Pressure and Cardiac Structure and Function? Int J Cardiovasc Sci, 2022. 35(2): p. 172-173.
  2. Kurtz, T.W. and R.C. Morris, Jr., Biological variability in Wistar-Kyoto rats. Implications for research with the spontaneously hypertensive rat. Hypertension, 1987. 10(1): p. 127-31.
  3. Rezende, L.M.T., et al., Is the Wistar Rat a more Suitable Normotensive Control for SHR to Test Blood Pressure and Cardiac Structure and Function? Int J Cardiovasc Sci, 2022. 35(2): p. 161-171.
  4. Moreira, N.J.D., et al., Acute renal denervation normalizes aortic function and decreases blood pressure in spontaneously hypertensive rats. Sci Rep, 2020. 10(1): p. 21826.
  5. Nakamura, A. and E.J. Johns, Renal nerves, renin, and angiotensinogen gene expression in spontaneously hypertensive rats. Hypertension, 1995. 25(4 Pt 1): p. 581-6.
  6. Shi, W., et al., Protective effect of calcitriol on podocytes in spontaneously hypertensive rat. J Chin Med Assoc, 2018. 81(8): p. 691-698.